# Machine learning-based on model for explain risk of 24-hour death in critically ill patients in the prehospital setting: A retrospective cohort study

Shengtao Li[1], Zhanzhan Li[2], Ruqiao Luo[1], Yanfen Li[1], Aoli Shi[1], Yanyan Li[3]*

1 Department of Emergency, the First Hospital of Changsha, the Affiliated Hospital of Changsha, Xiangya School of Medicine, Central South University, Changsha, Hunan Province, China, 2 Department of Oncology, Xiangya Hospital, Central South University, Changsha, Hunan Province, China, 3 Department of Nursing, Xiangya Hospital, Central South University, Changsha, Hunan Province, China

* liyanyan@csu.edu.cn

## Abstract

This study aimed to develop and validate a machine learning-based model for predicting 24-hour mortality in critically ill patients using prehospital and admission clinical data. We conducted a retrospective cohort study leveraging data from the prehospital emergency electronic medical record, in-hospital triage, and hospital information systems of a tertiary hospital in Changsha between August 2023 and April 2025. A total of 892 adult patients classified as critically ill were included. Nine machine learning algorithms were trained to predict 24-hour mortality, and model performance was assessed using the area under the receiver operating characteristic curve (AUC), sensitivity, specificity, accuracy, and F1 score. SHapley Additive exPlanations (SHAP) analysis was employed to interpret feature contributions. Among the nine algorithms, the Random Forest (RF) model exhibited the most stable and robust performance. Using nine selected features-prehospital heart rate, prehospital and admission systolic and diastolic blood pressure, prehospital and admission oxygen saturation, admission respiratory rate, and level of consciousness, the RF model achieved an AUC of 0.985(95%CI:0.976–0.993) in the training set and 0.863 (95%CI:0.766–0.961) in the testing set, demonstrating high accuracy and potential clinical applicability. SHAP analysis revealed that prehospital heart rate, admission respiratory rate, and blood pressure are the strongest predictors of mortality. Finally, the model was deployed as an interactive web-based tool for real-time clinical application. In summary, this study developed a simple, interpretable, and accurate machine learning model for predicting 24-hour mortality in critically ill prehospital patients. The RF-based model can be intended as an exploratory, hypothesis-generating tool and should supplement, not replace, clinical judgment. Further validation in larger, multi-center prospective cohorts with higher event rates is essential to confirm the robustness and real-world applicability of our findings.

**Data availability statement:** All relevant data are within the paper and its Supporting Information files.

**Funding:** The author(s) received no specific funding for this work.

**Competing interests:** The authors have declared that no competing interests exist.

## Introduction

Prehospital emergency care encompasses the comprehensive management of patients through urgent medical treatment and rapid transport prior to arrival at a healthcare facility. It plays a pivotal role in the management of life-threatening conditions such as cardiac arrest, metabolic disorders, respiratory failure, and severe trauma. The primary goal of prehospital emergency care is to ensure that patients receive timely and effective treatment and are transferred to the most appropriate medical institution. However, the spectrum of conditions encountered in prehospital emergencies is broad, with patients often presenting with complex, variable and rapidly evolving clinical manifestations. Many critically ill patients face life-threatening delays due to secondary transfers or inadequate clinical attention during transport. Therefore, early identification of "warning signs," appropriate triage, and prompt intervention are essential to prevent disease progression and improve patient outcomes. With ongoing advances in medical science, numerous scoring systems have been developed worldwide to assess disease onset, progression, and prognosis, thereby supporting clinical decision-making. Commonly used trauma assessment tools include the Abbreviated Injury Scale (AIS) [1], Injury Severity Score (ISS) [2], Revised Trauma Score (RTS) [3], Glasgow Coma Scale (GCS) [4], Trauma Score (TS) [5], CRAMS Scale [6], Prehospital Index (PHI) [7], Trauma Index (TI), and Triage Checklist (TC) [8]. Tools for evaluating disease severity and prognosis include the Early Warning Score (EWS), Acute Physiology and Chronic Health Evaluation II (APACHE II) [9], National Early Warning Score (NEWS) [10], Simplified Acute Physiology Score (SAPS) [11], Sequential Organ Failure Assessment (SOFA) [12], and Modified Early Warning Score (MEWS) [13]. Although these systems are well established and provide objective assessments, their reliance on multiple clinical parameters limits their practicality in prehospital settings, where simplicity, speed, and accuracy are paramount. Consequently, developing a feasible, rapid, and reliable prehospital scoring system remains a pressing challenge for emergency physicians.

A disease risk prediction model is a statistical or mathematical tool designed to estimate an individual's probability of developing a specific disease, thereby enabling early identification of potential health threats and facilitating timely clinical intervention to reduce morbidity and mortality [14]. These models typically integrate biological, environmental, and lifestyle factors with epidemiological data and medical knowledge to perform quantitative risk assessments. With the rapid advancement of artificial intelligence (AI), machine learning (ML) and deep learning (DL) algorithms have increasingly been applied in medicine, demonstrating exceptional performance in predictive modeling [15]. By integrating multidimensional information data including clinical variables, imaging features, radiotherapy dose parameters, and biomarkers, AI-based approaches can substantially improve the accuracy and applicability of disease risk prediction [16]. In this study, we utilized data from the prehospital emergency electronic medical record system, in-hospital triage system, and in-hospital electronic medical record database to develop a 24-hour mortality risk prediction model for critically ill prehospital patients using various machine learning algorithms. The optimal model was selected to provide a quantitative foundation for rational

allocation of prehospital emergency resources, stratified transfer decisions, and early intervention strategies, ultimately aiming to reduce early mortality and enhance the effectiveness of prehospital emergency care.

## Materials and methods

### Study population

This study followed the Transparent reporting of multivariable prediction models developed or validated using clustered data: TRIPOD-Cluster checklist Statement (S1 File). A retrospective cohort design was employed, including critically ill patients admitted to the emergency department via pre-hospital care at a tertiary hospital in Changsha between August 2023 and April 2025. Patients' data were accessed for research purposes on August 15, 2025. The inclusion criteria were as follows: (1) age > 18 years; (2) critically ill status, defined according to the guidelines for the classification of emergency patient conditions as "near death" or "critical" in the emergency triage system [17]; (3) admission via transfer from the medical emergency center; (4) complete electronic medical records; and (5) a clearly documented survival or death outcome within 24 hours. Exclusion criteria included: (1) patients declared dead before hospital arrival; (2) patients with incomplete clinical data; and (3) patients unable to cooperate with emergency treatment or those who abandoned treatment. Based on the principle of sample size estimation for multivariable regression analysis, the required sample size is generally a minimum of 10–20 cases per independent variable is generally recommended. Considering that 29 variables were planned for inclusion, the required sample size was estimated at 290–580 cases. Therefore, the actual inclusion of 892 patients was sufficient to meet the analytical requirements [18]. This study was approved by the Medical Ethics Committee of the First Hospital of Changsha (NO.: 2025−17).

### Data collection and processing

Patients' data were obtained from the prehospital emergency electronic medical record management system, the in-hospital triage system, and the hospital electronic medical record system. Collected variables included demographic characteristics (age and sex) and medical history, such as hypertension, coronary heart disease [CHD], diabetes, stroke, pulmonary disease, and hepatic or renal disease). Prehospital variables comprised transport time, pupillary light reflex status (pupils equal and round [PER], pupils reactive to light [PRL]), level of consciousness (conscious or unconscious), implementation of cardiopulmonary resuscitation (CPR), use of bag-mask ventilation and oxygen inhalation, body temperature (°C), heart rate (beats/min), respiratory rate (breaths/min), systolic blood pressure (SBP, mmHg), diastolic blood pressure (DBP, mmHg), oxygen saturation (%), and shock index (heart rate/SBP). In addition, vital signs were collected at hospital triage, including body temperature (°C), heart rate (beats/min), respiratory rate (breaths/min), SBP (mmHg), DBP (mmHg), oxygen saturation (%), and shock index (heart rate/SBP). The primary outcome of follow-up was survival status (survival or non-survival) within 24 hours after hospital admission. Outlier detection was performed for continuous data, and observations with clearly implausible values were excluded from the analysis.

### Establishment and validation of model

The study cohort was randomly divided into training and validation sets at an 8:2 ratio. Initially, the area under the curve (AUC) and 95% confidence intervals of 29 features were calculated for each of the 29 candidate features to assess their ability to predict 24-hour mortality. Features were ranked according to their AUC values and subsequently grouped into six subsets comprising 4, 9, 14, 19, 24, and 29 features, respectively. Nine machine learning algorithms were then employed to construct predictive models, including support vector machine (SVM), multilayer perceptron (MLP), random forest (RF), k-nearest neighbor (KNN), light gradient boosting machine (LightGBM), logistic regression (LR), eXtreme gradient boosting (XGBoost), elastic net (ENET), and decision tree (DT). To optimize model performance, hyperparameters were determined using a combination of grid search and manual fine-tuning. For each algorithm, model performance was evaluated across different feature subsets by calculating the sensitivity, specificity, positive predictive value (PPV), negative

predictive value (NPV), accuracy, and F1 score in both the training and validation sets. In addition, the 5-fold and 10-fold cross validation were performed to further assess model robustness and minimize the risk of overfitting.

### Feature selection and model explanation

The top five models ranked by AUC were selected for feature refinement. The original 29 features were subsequently divided into 12 subsets containing 2, 3, 4, 7, 8, 9, 10, 11, 14, 19, 24, and 29 features, and model performance metrics including AUC, sensitivity, specificity, PPV, NPV, accuracy, and F1 score were recalculated for each subset. For the selected ML model, features were progressively reduced until a marked decline in AUC was observed. The most stable model and parsimonious model was then identified and designated as the final predictive model. Model discrimination was evaluated using the area under the receiver operating characteristic curve (ROC–AUC), SHapley Additive exPlanations (SHAP) analysis was applied to interpret feature contributions and assess variable importance [19]. The flow of the study design was presented in Fig 1.

### Webpage deployment tool based on shinyAPP

To facilitate clinical applicability, the final predictive model was integrated into an interactive web application developed using ShinyApps (https://www.shinyapps.io/). By inputting the corresponding feature values required by the final model, the application generates the predicted probability of 24-hour mortality and provides an individualized mortality risk assessment for critically ill patients.

### Statistical analysis

All statistical analyses and figure generation were performed using R software (version 4.3.5) and GraphPad Prism 9.0. The original code can be accessible in S2 File. Continuous variables with a normal distribution were expressed as mean ± standard deviation (x̄ ± s) and compared between groups using the independent samples t-test. Variables with a non-normal distribution were expressed as median and interquartile range [M (P25, P75)], with group comparisons conducted using the Mann–Whitney U test. Categorical variables were presented as frequencies and percentages, and comparisons between groups were performed using the chi-square test or Fisher's exact test, as appropriate. The receiver operating characteristics curve was used to predict the ability of features and models. The calibration plots and decision curve analysis (DCA) were used to evaluate model calibration and clinical utility. Restricted cubic spline models were employed to explore the dose–response relationship between continuous variables and 24-hour mortality risk in critically ill patients. Unless otherwise specified, a two-sided P value <0.05 was considered statistically significant.

### Ethics statement

This study was approved by the Medical Ethics Committee of the First Hospital of Changsha (NO.: 2025−17), all data were fully anonymized before we accessed them, and ethics committee waived the requirement for informed consent due to the retrospective nature of the study.

## Results

### Characteristics of study population

Table 1 presents the baseline characteristics of the study population and comparisons between training and test sets, as well as between survival and non-survival groups. A total of 892 patients were included, of whom 714 were allocated to the training set and 178 to the test set; no statistically significant differences were observed between the two sets, indicating good comparability. Overall, 65.1% of patients were male, with a mean age of 67.5 ± 18.0 years. The prevalence of hypertension, CHD, diabetes, stroke, lung disease, and liver/kidney disease were 53.5%, 72.9%, 80.4%, 88.5%, 96.2%, and 93.5%, respectively. The mean prehospital transport time was 20.3 ± 10.1 minutes.

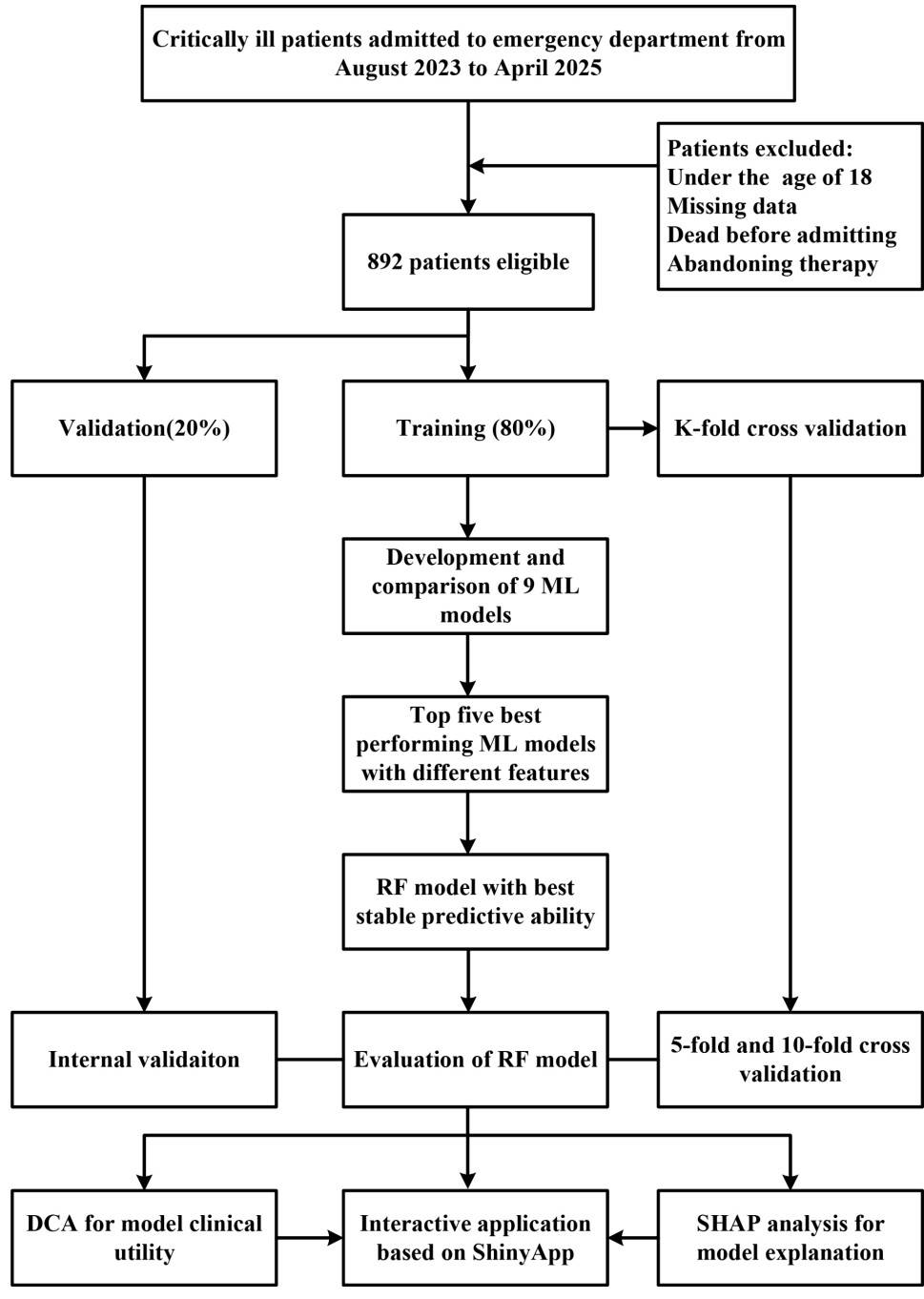

**Fig 1. Flow chart of the study design.**

In the training set, 51 patients (12.2%) died within 24 hours of admission. Compared with survivors, non-survivors showed no statistically significant differences in age, sex, comorbidities, or transport time (all P > 0.05). However, non-survivors exhibited markedly higher proportions of non-PER (94.1% vs. 60.8%, P < 0.001), non-PRL (96.8% vs. 52.9%, P < 0.001), and unconsciousness (67.0% vs. 9.8%, P < 0.001), and were less likely to receive prehospital CPR or BMV (both P < 0.001). Regarding physiological parameters, non-survivors showed significantly higher prehospital temperature,

**Table 1. Comparison of clinical characteristics between training and test sets, and between survival and non-survival groups.**

| Variables | Total (n = 892) | Test (n = 178) | Training (n = 714) | P | Non-survival (n = 51) | Survival (n = 663) | P |
|---|---|---|---|---|---|---|---|
| Age, year | 67.47 ± 17.96 | 66.54 ± 17.30 | 67.71 ± 18.12 | 0.438 | 72.08 ± 15.71 | 67.37 ± 18.26 | 0.074 |
| Gender, n (%) | | | | 0.992 | | | 0.327 |
| Male | 581 (65.13) | 116 (65.17) | 465 (65.13) | | 30 (58.82) | 435 (65.61) | |
| Female | 311 (34.87) | 62 (34.83) | 249 (34.87) | | 21 (41.18) | 228 (34.39) | |
| Hypertension, n (%) | | | | 0.761 | | | 0.088 |
| No | 415 (46.52) | 81 (45.51) | 334 (46.78) | | 18 (35.29) | 316 (47.66) | |
| Yes | 477 (53.48) | 97 (54.49) | 380 (53.22) | | 33 (64.71) | 347 (52.34) | |
| CHD, n (%) | | | | 0.666 | | | 0.104 |
| No | 242 (27.13) | 46 (25.84) | 196 (27.45) | | 9 (17.65) | 187 (28.21) | |
| Yes | 650 (72.87) | 132 (74.16) | 518 (72.55) | | 42 (82.35) | 476 (71.79) | |
| Diabetes, n (%) | | | | 0.538 | | | 0.312 |
| No | 175 (19.62) | 32 (17.98) | 143 (20.03) | | 13 (25.49) | 130 (19.61) | |
| Yes | 717 (80.38) | 146 (82.02) | 571 (79.97) | | 38 (74.51) | 533 (80.39) | |
| Stroke, n (%) | | | | 0.153 | | | 0.482 |
| No | 103 (11.55) | 26 (14.61) | 77 (10.78) | | 4 (7.84) | 73 (11.01) | |
| Yes | 789 (88.45) | 152 (85.39) | 637 (89.22) | | 47 (92.16) | 590 (88.99) | |
| Lung diseases, n (%) | | | | 0.067 | | | 0.062 |
| No | 123 (13.79) | 17 (9.55) | 106 (14.85) | | 3 (5.88) | 103 (15.54) | |
| Yes | 769 (86.21) | 161 (90.45) | 608 (85.15) | | 48 (94.12) | 560 (84.46) | |
| Liver kidney diseases, n (%) | | | | 0.845 | | | 0.276 |
| No | 58 (6.50) | 11 (6.18) | 47 (6.58) | | 1 (1.96) | 46 (6.94) | |
| Yes | 834 (93.50) | 167 (93.82) | 667 (93.42) | | 50 (98.04) | 617 (93.06) | |
| Transit time, minute | 20.26 ± 10.12 | 20.49 ± 10.72 | 20.20 ± 9.97 | 0.733 | 17.77 ± 8.19 | 20.39 ± 10.08 | 0.072 |
| PER, n (%) | | | | 0.502 | | | <0.001 |
| No | 821 (92.04) | 166 (93.26) | 655 (91.74) | | 31 (60.78) | 624 (94.12) | |
| Yes | 71 (7.96) | 12 (6.74) | 59 (8.26) | | 20 (39.22) | 39 (5.88) | |
| PRL, n (%) | | | | 0.453 | | | <0.001 |
| No | 833 (93.39) | 164 (92.13) | 669 (93.70) | | 27 (52.94) | 642 (96.83) | |
| Yes | 59 (6.61) | 14 (7.87) | 45 (6.30) | | 24 (47.06) | 21 (3.17) | |
| Awareness, n (%) | | | | 0.325 | | | <0.001 |
| Unconscious | 568 (63.68) | 119 (66.85) | 449 (62.89) | | 5 (9.80) | 444 (66.97) | |
| Conscious | 324 (36.32) | 59 (33.15) | 265 (37.11) | | 46 (90.20) | 219 (33.03) | |
| CPR, n (%) | | | | 0.175 | | | <0.001 |
| No | 47 (5.27) | 13 (7.30) | 34 (4.76) | | 23 (45.10) | 11 (1.66) | |
| Yes | 845 (94.73) | 165 (92.70) | 680 (95.24) | | 28 (54.90) | 652 (98.34) | |
| BMV, n (%) | | | | 0.661 | | | <0.001 |
| No | 73 (8.18) | 16 (8.99) | 57 (7.98) | | 27 (52.94) | 30 (4.52) | |
| Yes | 819 (91.82) | 162 (91.01) | 657 (92.02) | | 24 (47.06) | 633 (95.48) | |
| Oxygen inhalation, n (%) | | | | 0.376 | | | 0.058 |
| No | 542 (60.76) | 103 (57.87) | 439 (61.48) | | 25 (49.02) | 414 (62.44) | |
| Yes | 350 (39.24) | 75 (42.13) | 275 (38.52) | | 26 (50.98) | 249 (37.56) | |
| Prehospital | | | | | | | |
| Temperature, °C | 36.72 ± 0.80 | 36.75 ± 0.74 | 36.72 ± 0.81 | 0.579 | 36.35 ± 1.09 | 36.74 ± 0.78 | 0.015 |
| Heart Rate, n | 93.31 ± 28.50 | 94.43 ± 29.48 | 93.03 ± 28.26 | 0.557 | 63.04 ± 41.16 | 95.33 ± 25.65 | <0.001 |
| Respiratory frequency, n | 20.52 ± 5.77 | 20.65 ± 5.29 | 20.48 ± 5.88 | 0.725 | 13.29 ± 10.07 | 21.03 ± 5.03 | <0.001 |

*(Continued)*

**Table 1.** (Continued)

| Variables | Total (n = 892) | Test (n = 178) | Training (n = 714) | P | Non-survival (n = 51) | Survival (n = 663) | P |
|---|---|---|---|---|---|---|---|
| SBP, mmHg | 134.67 ± 46.85 | 138.37 ± 47.51 | 133.74 ± 46.67 | 0.239 | 77.71 ± 46.38 | 138.05 ± 43.85 | <0.001 |
| DBP, mmHg | 80.11 ± 28.66 | 82.54 ± 28.30 | 79.50 ± 28.74 | 0.206 | 47.45 ± 31.88 | 81.97 ± 26.97 | <0.001 |
| Blood oxygen, % | 89.40 ± 11.60 | 90.10 ± 10.95 | 89.23 ± 11.75 | 0.374 | 72.84 ± 15.38 | 90.49 ± 10.42 | <0.001 |
| Shock index | 0.79 ± 0.44 | 0.76 ± 0.36 | 0.79 ± 0.46 | 0.374 | 0.89 ± 0.55 | 0.79 ± 0.45 | 0.117 |
| Admission | | | | | | | |
| Temperature, °C | 37.80 ± 11.74 | 37.74 ± 11.49 | 37.82 ± 11.81 | 0.940 | 36.23 ± 1.33 | 37.94 ± 12.24 | 0.320 |
| Heart rate, n | 91.70 ± 28.85 | 92.73 ± 28.96 | 91.45 ± 28.84 | 0.596 | 62.00 ± 39.32 | 93.71 ± 26.59 | <0.001 |
| Respiratory frequency, n | 20.39 ± 9.14 | 20.20 ± 10.39 | 20.44 ± 8.81 | 0.747 | 12.18 ± 9.71 | 21.08 ± 8.42 | <0.001 |
| SBP, mmHg | 135.23 ± 45.44 | 136.94 ± 46.70 | 134.81 ± 45.15 | 0.575 | 75.27 ± 44.52 | 139.39 ± 41.85 | <0.001 |
| DBP, mmHg | 77.35 ± 26.61 | 79.36 ± 27.44 | 76.85 ± 26.40 | 0.260 | 42.80 ± 26.38 | 79.47 ± 24.53 | <0.001 |
| Blood oxygen, % | 90.53 ± 12.39 | 91.76 ± 12.58 | 90.22 ± 12.33 | 0.139 | 72.61 ± 15.55 | 91.58 ± 10.94 | <0.001 |
| Shock index | 0.74 ± 0.32 | 0.75 ± 0.36 | 0.74 ± 0.31 | 0.706 | 0.85 ± 0.33 | 0.73 ± 0.30 | 0.006 |

BMV: Bag-mask ventilation; CPR: cardio-pulmonary resuscitation; CHD: coronary heart disease; DBP: diastolic blood pressure. PER: pupils equal and round; PRL: pupils reactive to light; SBP: Systolic blood pressure.

heart rate, respiratory rate, SBP, DBP, and oxygen saturation (all P < 0.05), with no difference in shock index. At admission, non-survivors continued to exhibit higher heart rate, respiratory rate, blood pressure, and oxygen saturation, whereas the shock index was statically significantly lower than in survivors (P = 0.006).

## Does-response relationship

Based on the univariate comparisons between the survival and non-survival groups, we further investigated the non-linear associations between prehospital and admission physiological parameters and 24-hour mortality risk in critically ill patients. Variables demonstrating statistically significant differences between groups were included in this analysis, including body temperature, heart rate, respiratory rate, systolic blood pressure (SBP), diastolic blood pressure (DBP), and shock index. The dose–response curves (Fig 2A-2N) demonstrated nonlinear relationships between prehospital and admission temperature, heart rate, respiratory rate, prehospital SBP, DBP, and admission shock index and the risk of mortality (P < 0.05 for overall association; P < 0.05 for nonlinearity). Specifically, the mortality risk increased when prehospital or admission temperature exceeded 36.2°C or when heart rate was greater than 60 beats per minute. Conversely, mortality risk decreased rapidly when prehospital respiratory rate was greater than 4 breaths per minute or admission respiratory rate exceeded 15, as well as when prehospital SBP was above 74 mmHg, DBP above 44.3 mmHg, or admission shock index below 0.69. For prehospital and admission oxygen saturation, admission SBP and DBP, and prehospital shock index, statistically significant linear associations with mortality risk were observed (P < 0.05 for overall association; P > 0.05 for nonlinearity). Mortality risk decreased when prehospital or admission oxygen saturation exceeded 70%, admission SBP was greater than 79.3 mmHg or DBP exceeded 42 mmHg, or when prehospital shock index was below 0. 69.

## Model development and performance comparison

Using data collected from both the prehospital and admission phases, nine machine learning models were developed to predict 24-hour mortality risk in critically ill patients based on 29 candidate features. In the training set, the support vector machine (SVM) model achieved the highest discriminative performance (AUC = 0.998), followed by the multilayer perception (MLP; AUC = 0.993) and random forest (RF; AUC = 0.986) models. The k-nearest neighbor (KNN; AUC = 0.961), Light Gradient Boosting Machine (LightGBM; AUC = 0.959), and logistic regression (LR; AUC = 0.935) models demonstrated

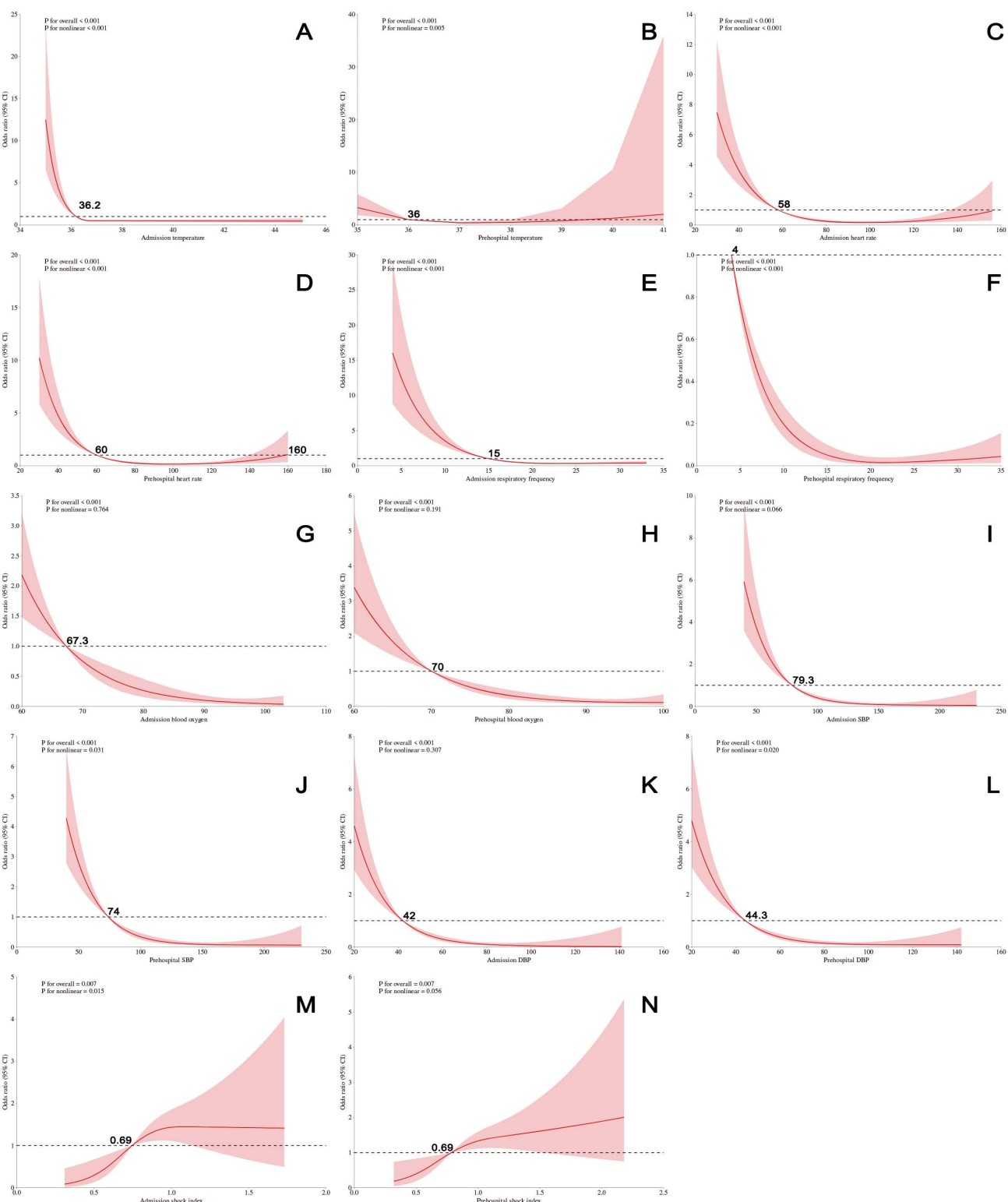

**Fig 2. Restricted cubic spline plots illustrate the relationships between different continuous variables and 24-hour mortality risk. A and B:** Admission and prehospital temperature; **C and D:** Admission and prehospital heart rate; **E and F:** Admission and prehospital respiratory frequency; **G and H:** Admission and prehospital blood oxygen; **I and J:** Admission and prehospital SBP; **K and L:** Admission and prehospital DBP; **M and N:** Admission and prehospital shock index.

comparable discriminative ability, whereas the eXtreme Gradient Boosting (XGBoost), elastic net (ENET), and decision tree (DT) models yielded AUC values below 0.900. In the testing set, the elastic net (ENET) model showed the highest discriminative performance (AUC = 0.904), followed by the MLP (AUC = 0.895), logistic regression (LR; AUC = 0.891), and XGBoost (AUC = 0.887) models. The SVM, RF, and LightGBM models exhibited similar performances, with AUC values ranging from 0.860 to 0.865. In contrast, the KNN (AUC = 0.797) and DT (AUC = 0.789) models demonstrated relatively low predictive performance. Detailed performance metrics, including sensitivity, specificity, positive predictive value (PPV), negative predictive value (NPV), accuracy, and F1 score for each model, are summarized in Table 2.

## Identification and validation of final model

To identify the optimal model, the top five models in the training set (SVM, MLP, RF, KNN, and LightGBM) were selected for further optimization based on their AUC values. The 29 features were categorized into six groups according to the AUC of each single variable (S1 Table), and each of the five models was reconstructed using 4, 9, 14, 19, 24, and 29 features, respectively. During the feature reduction process, the RF model exhibited the most stable predictive performance across all six feature groups (AUC = 0.981–0.986, Fig 3A). In the training set, the RF model achieved the highest predictive accuracy (AUC = 0.985, 95%CI:0.976–0.993)). When nine features were included (prehospital heart rate, SBP, DBP, blood oxygen, and admission SBP, DBP, and blood oxygen, and awareness), the model also demonstrated good calibration and the smallest deviation in predicted probabilities (S2 Table). DCA further indicated that the RF model achieved the greatest net clinical benefit across the entire threshold range (0.0–0.8), followed by the LightGBM model. In the testing set, the RF model achieved an AUC of 0.863 when built with nine features (Fig 3B), ranking third among the five candidate models. Consistently, calibration curves and DCA results confirmed that the RF model maintained superior model fitting and clinical utility (S1 Fig).

**Table 2. Performance of the ML models for mortality risk prediction based on all features.**

| Dataset | Model | AUC | Sensitivity | Specificity | PPV | NPV | Accuracy | F1 score |
|---|---|---|---|---|---|---|---|---|
| Training | SVM | 0.998 | 0.997 | 1.000 | 1.000 | 0.952 | 0.997 | 0.998 |
| | MLP | 0.993 | 0.967 | 0.975 | 0.998 | 0.672 | 0.968 | 0.983 |
| | RF | 0.986 | 0.955 | 1.000 | 1.000 | 0.606 | 0.958 | 0.977 |
| | KNN | 0.961 | 0.796 | 1.000 | 1.000 | 0.252 | 0.809 | 0.887 |
| | LightGBM | 0.959 | 0.930 | 0.900 | 0.993 | 0.468 | 0.928 | 0.960 |
| | LR | 0.935 | 0.889 | 0.875 | 0.990 | 0.350 | 0.888 | 0.937 |
| | XGboost | 0.898 | 0.899 | 0.800 | 0.985 | 0.352 | 0.893 | 0.940 |
| | ENET | 0.890 | 0.902 | 0.775 | 0.983 | 0.352 | 0.894 | 0.941 |
| | DT | 0.731 | 0.475 | 0.986 | 0.965 | 0.704 | 0.954 | 0.975 |
| Testing | SVM | 0.860 | 0.959 | 0.545 | 0.959 | 0.545 | 0.925 | 0.959 |
| | MLP | 0.895 | 0.951 | 0.636 | 0.967 | 0.538 | 0.925 | 0.959 |
| | RF | 0.868 | 0.947 | 0.591 | 0.963 | 0.500 | 0.918 | 0.955 |
| | KNN | 0.797 | 0.780 | 0.682 | 0.950 | 0.217 | 0.772 | 0.863 |
| | LightGBM | 0.860 | 0.947 | 0.636 | 0.967 | 0.519 | 0.922 | 0.957 |
| | LR | 0.891 | 0.919 | 0.682 | 0.970 | 0.429 | 0.899 | 0.944 |
| | XGboost | 0.887 | 0.931 | 0.682 | 0.970 | 0.469 | 0.910 | 0.950 |
| | ENET | 0.904 | 0.939 | 0.636 | 0.967 | 0.483 | 0.914 | 0.953 |
| | DT | 0.789 | 0.988 | 0.591 | 0.964 | 0.812 | 0.955 | 0.976 |

AUC: area under the curve; DT: decision; ENET: Elastic net; tree; KNN: K-nearest neighbor; LightGBM: light gradient boosting machine; LR: logistic regression; MAL: machine learning; MLP: Multilayer perception; NPV: negative predictive value; PPV: positive predictive value; RF: random forest; SVM: support vector machine; XGboost: eXtreme gradient boosting;

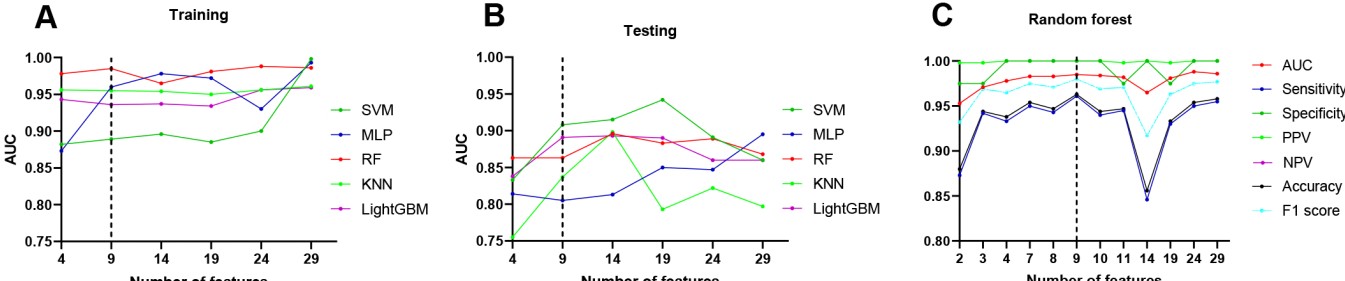

**Fig 3. Performance of machine learning models to predict 24-hour mortality risk. A**: AUCs of the top five machine learning models with varied number of features in training dataset. **B**: AUCs of the top five machine learning models with varied number of features in testing dataset. **C**: AUC, sensitivity, specificity, PPV, NPV, accuracy and F1 score of random forest model with varied number of features.

We further compared the discriminative performance of the random forest (RF) model constructed using different feature sets. For each configuration, sensitivity, specificity, positive predictive value (PPV), negative predictive value (NPV), accuracy, and F1 score were calculated. The RF model incorporating nine features demonstrated the most favorable overall balance across performance metrics (Fig 3C). In the training set, this model achieved a sensitivity of 0.961, specificity of 1.000, PPV of 1.000, NPV of 0.635, accuracy of 0.963, and an F1 score of 0.980. In the testing set, the corresponding values were a sensitivity of 0.943, specificity of 0.636, PPV of 0.967, NPV of 0.500, accuracy of 0.918, and an F1 score of 0.955. Detailed performance metrics for all feature configurations are provided in S2 Table. Based on these results, the RF model incorporating nine features—prehospital heart rate; admission respiratory rate; prehospital and admission SBP; prehospital and admission DBP; prehospital and admission oxygen saturation; and level of consciousness—was selected as the final predictive model for mortality risk in critically ill patients. To further assess model robustness, both 5-fold and 10-fold cross-validation were performed. The mean AUCs were 0.881 (95% CI: 0.789–0.942) for 5-fold cross-validation and 0.874 (95% CI: 0.786–0.931) for 10-fold cross-validation, indicating stable and consistent model performance.

## Model explanation

Because clinicians often find it difficult to accept predictive models that are either poorly interpretable or entirely opaque, we employed the SHAP method to interpret the output of the final model by quantifying the contribution of each variable to the prediction. This explainable approach provides two levels of interpretability: a global interpretation at the feature level and a local interpretation at the individual level. The global interpretation describes the overall behavior of the model. As shown in the SHAP summary bar plot (Fig 4A), the mean SHAP values were used to evaluate the contribution of each feature to the model output, arranged in descending order of importance. The nine most influential features in the predictive model were prehospital heart rate, admission respiratory rate, prehospital SBP, admission blood oxygen saturation, admission SBP, admission DBP, prehospital DBP, awareness status, and prehospital blood oxygen saturation. Furthermore, the SHAP summary dot plot (Fig 4B) visually illustrates both the direction and magnitude of each feature's influence on model output. Features such as prehospital systolic blood pressure (SBP), blood oxygen saturation, and admission SBP showed notable contributions to the model output, with higher values generally corresponding to SHAP values in the direction of lower predicted mortality. The SHAP dependence plots illustrate how variations in individual feature values are associated with changes in the model's prediction. The SHAP waterfall plot (Fig 4C) presents the contribution of each feature to the mortality prediction for a representative critically ill patient. Specifically, the feature values and their corresponding SHAP scores indicate whether a given feature shifts the model output toward higher or lower predicted mortality. For example, in a patient with a prehospital SBP of 72 mmHg and an admission oxygen saturation of 85%, the corresponding SHAP values were −0.0162 and −0.0131, respectively, both shifting the model output toward a lower predicted mortality.

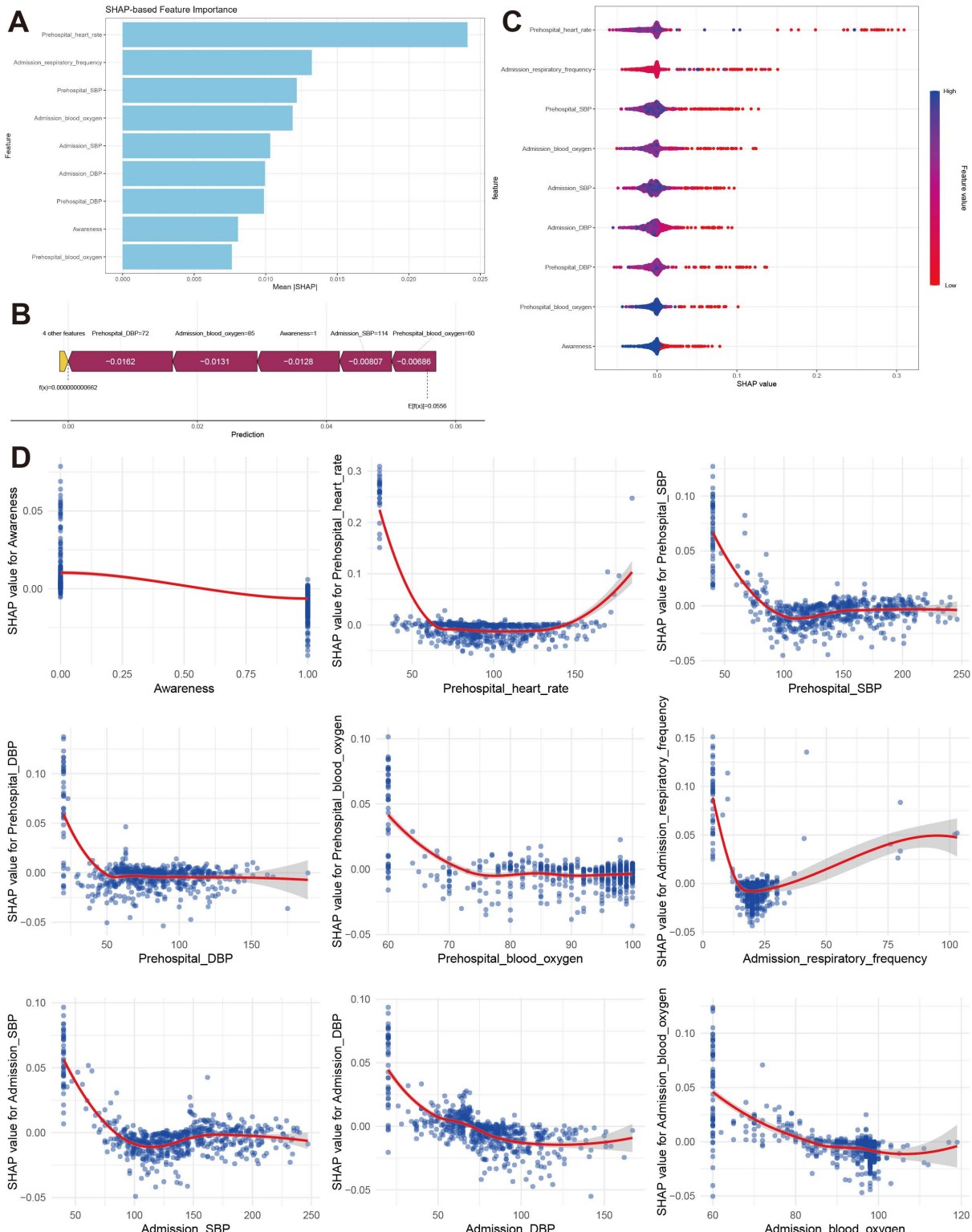

**Fig 4. Shapley Additive exPlanation analysis for global and local model explanations based on random forest model. A**: Evaluation of feature importance based on SHAP mean values, presented in descending order; **B**: Force plot shows the effect of each feature on prediction outcome. **C**: SHAP summary dot plot displayed that the probability of 24-hour mortality increased with the SHAP values of these features. **D**: SHAP dependence plot shows the effect of s single feature on model's output. Each point represents a patient.

Furthermore, Fig 4D compares the actual values and SHAP values of the nine key features. A SHAP value greater than zero corresponds to a shift of the model output toward the "survival" prediction, whereas a SHAP value less than zero indicates a shift toward higher predicted mortality. For instance, higher prehospital SBP (>72 mmHg), higher admission oxygen saturation (>85%), a consciousness score of 1, higher admission diastolic blood pressure (>114 mmHg), and higher prehospital oxygen saturation (>60%) were generally associated with positive SHAP values, indicating a greater contribution to survival-oriented model predictions.

### Interactive application of model

To enhance the practicality and usability of the model, we integrated it into a web-based application to assist clinicians in more accurately assessing mortality risk in critically ill patients (Fig 5). By entering the specific values of the nine selected features, this ShinyApp-based tool automatically calculates the predicted probability of death and classifies patients into corresponding risk categories based on the estimated probability. The web application is publicly accessible at the following link: https://liyanyan.shinyapps.io/RandomForest/.

### Discussion

To our knowledge, this is the first study to systematically investigate and compare nine machine learning models for predicting 24-hour mortality risk among critically ill patients in the prehospital setting. We identified a set of risk factors associated with early mortality and developed and validated a predictive model using multiple machine learning algorithms. Machine learning represents a powerful computational approach for analyzing complex, high-dimensional data and for flexibly capturing nonlinear and interactive relationships among variables [20]., making it particularly well suited for predictive modeling in clinical medicine.. Among the nine models evaluated, the random forest model constructed using nine selected features demonstrated the most favorable overall predictive performance. Random forest is an ensemble learning algorithm that aggregates multiple decision trees generated through random sampling of both observations and features, with final predictions determined by majority voting [21]. This framework confers several advantages, including high predictive accuracy, robustness to noise, reduced susceptibility to overfitting, and the capacity to quantify feature importance, thereby facilitating the identification of key clinical predictors [22]. Previous studies have consistently shown the strong predictive capability of machine learning algorithms across a wide range of medical applications [23,24]. Notably, machine learning does not impose strict assumptions regarding feature selection or the number of variables included in a model. Consequently, an important methodological consideration is achieving an appropriate balance between clinical applicability and predictive performance, particularly when the inclusion of non-causal or redundant features may reduce model interpretability or generalizability. In the present study, we addressed this challenge by adopting a feature ranking strategy based on optimal AUC performance. This approach enabled the development of a parsimonious predictive model incorporating a limited number of clinically accessible variables while preserving strong discriminative ability, thereby enhancing its potential utility in real-world clinical decision-making for critically ill patients..

In this study, the random forest model constructed using prehospital data incorporated nine easily obtainable variables: prehospital heart rate, SBP, DBP, blood oxygen saturation, admission SBP, DBP and respiratory frequency, blood oxygen saturation, and level of consciousness. These features can be readily acquired during prehospital emergency care. In prehospital emergency care, heart rate, respiratory rate, oxygen saturation, and blood pressure are key vital signs used to assess the severity of critical illness [25]. The SHAP analysis revealed that prehospital heart rate ranked first in predictive importance. For critically ill patients, a heart rate below 40 beats per minute (bpm) or above 120 bpm typically indicates a high-risk condition. Bradycardia (<40 bpm) may be associated with hypothermia, electrolyte imbalance, or acid–base disturbances, which not only hinder recovery but may also exacerbate the disease [26]. Conversely, a persistent tachycardia (>120 bpm) increases myocardial workload and oxygen consumption, thereby accelerating clinical deterioration [27]. The dose–response analysis in this study indicated that when the prehospital heart rate was below 60 bpm, the 24-hour

# Critically Ill Patients Mortality Risk Predictor

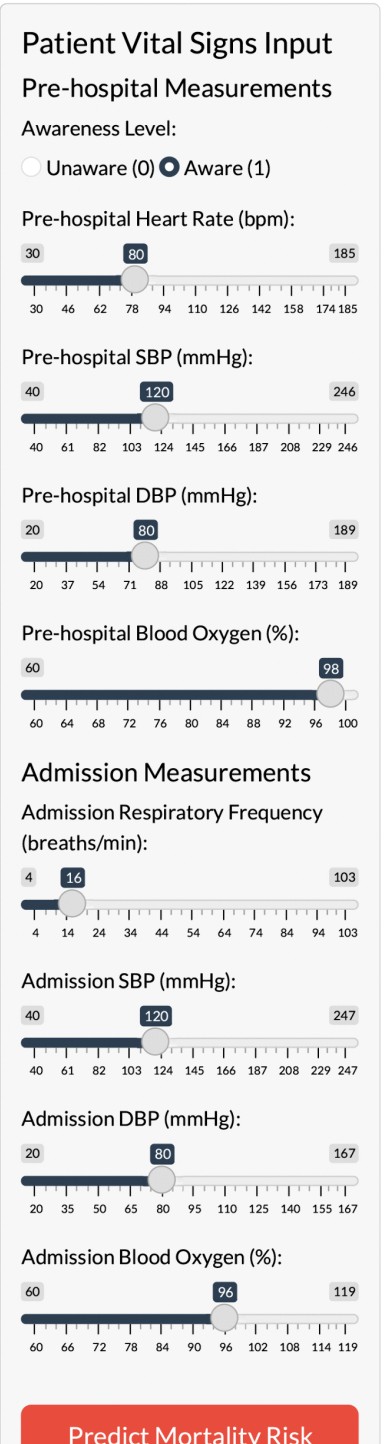

**Patient Vital Signs Input**

Pre-hospital Measurements

Awareness Level:

○ Unaware (0)  ● Aware (1)

Pre-hospital Heart Rate (bpm):

30 — [80] — 185
30  46  62  78  94  110  126  142  158  174 185

Pre-hospital SBP (mmHg):

40 — [120] — 246
40  61  82  103  124  145  166  187  208  229 246

Pre-hospital DBP (mmHg):

20 — [80] — 189
20  37  54  71  88  105  122  139  156  173 189

Pre-hospital Blood Oxygen (%):

60 — [98] — 100
60  64  68  72  76  80  84  88  92  96  100

**Admission Measurements**

Admission Respiratory Frequency (breaths/min):

4 — [16] — 103
4  14  24  34  44  54  64  74  84  94  103

Admission SBP (mmHg):

40 — [120] — 247
40  61  82  103  124  145  166  187  208  229 247

Admission DBP (mmHg):

20 — [80] — 167
20  35  50  65  80  95  110  125  140  155 167

Admission Blood Oxygen (%):

60 — [96] — 119
60  66  72  78  84  90  96  102  108  114 119

[ Predict Mortality Risk ]

## Mortality Risk Assessment

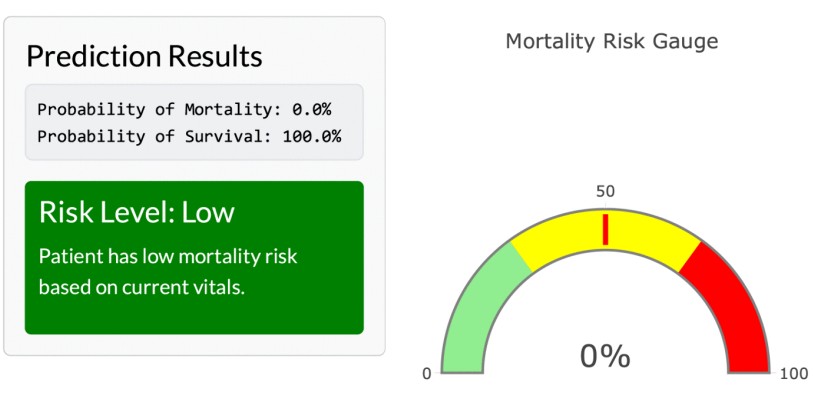

**Prediction Results**

```
Probability of Mortality: 0.0%
Probability of Survival: 100.0%
```

**Risk Level: Low**

Patient has low mortality risk based on current vitals.

Mortality Risk Gauge

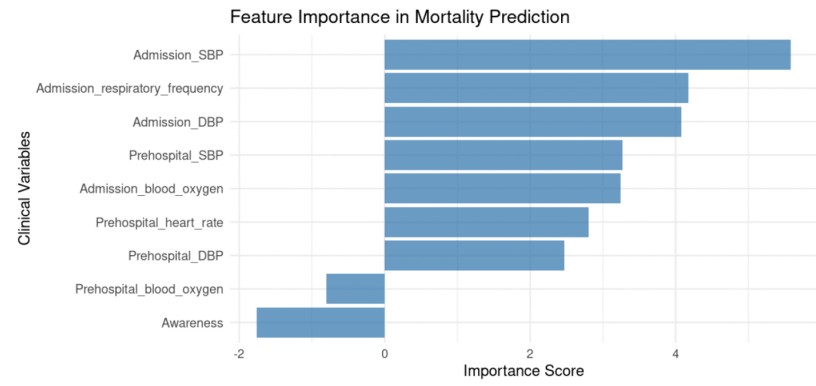

**Fig 5. Interactive application of random forest model.** This interactive application based on random forest model with nine features is available for predicting 24-hour mortality risk of critically ill patients. This interactive web application can give the probability of mortality when actual values of nine features were entered. Meanwhile, it also gives risk level and feature importance.

mortality risk increased; however, no statistically significant nonlinear association was observed when the prehospital heart rate exceeded 120 bpm. Abnormal respiratory frequencies are likewise life-threatening. In healthy adults at rest, the normal respiratory rate ranges from 12 to 20 breaths per minute. A sustained rate exceeding 30 breaths per minute, dropping below 8 breaths per minute, or episodes of apnea lasting longer than one minute are often indicative of severe illness or organ failure [28]. The dose–response curves in this study further demonstrated a nonlinear association between admission respiratory rate and 24-hour mortality risk in critically ill patients, with mortality risk increasing sharply when the respiratory rate fell below 15 breaths per minute. While mild hypotension alone typically does not result in rapid deterioration, severe, acute-onset hypotension—or hypotension occurring in the context of specific comorbidities or major complications—can pose a substantial threat to patient survival. [29]. For instance, a sudden drop in systolic blood pressure to extremely low levels can result in inadequate perfusion of vital organs such as the heart, brain, and kidneys, leading to arrhythmia, syncope, shock, or even sudden death [30]. In the context of severe cardiac disease, massive hemorrhage, serious infection, or anaphylaxis, hypotension can progress rapidly and become life-threatening if not promptly managed [31]. Given the prevalence of these conditions among critically ill patients, blood pressure is closely linked to mortality risk. In the predictive model developed in this study, both prehospital and admission SBP and DBP were included, further underscoring the prognostic significance of blood pressure in this population. Oxygen saturation is a critical indicator of respiratory and circulatory function in emergency patients, with a normal reference range of 95%–99%. Levels below 90% indicate hypoxemia, with lower values reflecting progressively severe oxygen deprivation. Without timely intervention, hypoxemia significantly increases the risk of death. [32]. Moreover, blood oxygen is closely correlated with respiratory rate, and critically ill patients often develop a vicious cycle between respiratory failure and hypoxemia [33]. This study also identified both prehospital and admission blood oxygen saturation as crucial predictors of mortality in critically ill patients. Level of consciousness is another important determinant of prognosis in this population [34]. Delirium, an acute and fluctuating disturbance of cognitive function often accompanied by hallucinations or illusions, is common among critically ill patients. Clinically, delirium is categorized into hyperactive and hypoactive subtypes. The hypoactive form, which presents lethargy and decreased responsiveness, is easily overlooked and associated with higher mortality [35]. In contrast, hyperactive patients typically exhibit agitation and restlessness, with relatively lower mortality and a greater likelihood of recovery [36]. Delirium not only reflects acute cerebral dysfunction but is also linked to long-term and irreversible neurological damage when it persists [37]. Our study based on local explanation from the SHAP analysis found that once altered consciousness occurred, the patient's risk of death increased significantly.

Our model demonstrated excellent predictive performance in both internal validation and cross-validation, achieving an AUC of 0.863 (95% CI: 0.766–0.961) in the internal validation dataset, with mean AUCs of 0.881 and 0.874 in the 5-fold and 10-fold cross-validations, respectively. To enhance interpretability, we applied the SHapley Additive exPlanations (SHAP) method, allowing clinicians to better understand the model's predictions and improve clinical efficiency. By providing explanations for the machine learning model, SHAP addresses the common challenge of clinician hesitation arising from the "black-box" nature of traditional predictive algorithms. It offers both global interpretations, which describe the overall behavior of the model, and individualized explanations, which illustrate how a patient's specific input data contributes to their predicted risk. Furthermore, by inputting relevant clinical data into the web-based application developed in this study, clinicians can rapidly estimate a patient's 24-hour mortality risk, thereby supporting timely and informed decision-making in the prehospital emergency setting.Despite its findings, this study has several limitations that warrant cautious interpretation. First, the inherent constraints of the dataset pose a challenge to model stability. Although the total cohort included 892 patients, the low number of outcome events (n = 51) resulted in a critically low events-per-variable (EPV) ratio. While we employed feature selection and regularization to mitigate these risks, such a low EPV may still lead to model instability and potential overfitting. The observed performance gap between the training and test sets further highlights the risk of "optimism" in predictive accuracy. Additionally, the limited event rate precluded detailed subgroup analyses, which is significant given the high heterogeneity of critically ill patients [38]. Second, the study's

scope was restricted by the temporal and categorical nature of the data. Our model relied exclusively on prehospital and on-admission physiological parameters within a 24-hour window, omitting clinical biochemical indicators and the dynamic temporal trajectories of illness during ICU hospitalization. Furthermore, the analysis focused solely on short-term (24-hour) mortality; thus, the model's utility in predicting long-term outcomes, such as 30-day or 180-day mortality, remains unexamined. Third, the generalizability and causal interpretation of the model are limited. As a retrospective study conducted at a single center without external validation, the findings may not readily generalize to diverse clinical populations or varying institutional practices. Moreover, the identified predictors represent statistical correlations rather than causal links. Future research should employ formal causal inference frameworks—such as target trial emulation or inverse probability weighting—to evaluate whether these factors are truly modifiable targets for clinical intervention [39]. Finally, this model is intended as an exploratory, hypothesis-generating tool and should supplement, not replace, clinical judgment. Further validation in larger, multi-center prospective cohorts with higher event rates is essential to confirm the robustness and real-world applicability of our findings.

## Conclusion

In conclusion, we developed an interpretable machine learning–based model for predicting 24-hour mortality risk among critically ill patients in the prehospital setting. The model was constructed using readily available clinical variables obtained from the prehospital emergency electronic medical record system, in-hospital triage system, and hospital electronic medical record database. The final random forest (RF) model demonstrated strong discriminative performance for early mortality during internal validation. This model offers a quantitative tool to assist emergency medical personnel in the rapid identification of high-risk patients prior to hospital transport, supports timely and individualized clinical decision-making, and facilitates optimized allocation of prehospital emergency resources and stratified transfer strategies. Collectively, these advantages have the potential to improve the efficiency of prehospital emergency care and reduce early mortality among critically ill patients. Future studies are warranted to externally validate and further refine this model using larger, multicenter, and multi-regional cohorts, thereby enhancing its robustness and generalizability.

## Supporting information

**S1 File. TRIPOD checklist.**
(DOCX)

**S2 File. Original code.**
(TXT)

**S1 Table. The predictive ability of single parameter for mortality risk in critically ill patients.**
(DOCX)

**S2 Table. Performance of the top five ML models for mortality risk prediction based on nine features.**
(DOCX)

**S1 Fig. Performance of the top five models predicting 24-hour mortality risk in training and testing datasets. A and B:** The receiver operating characteristics curve with AUCs; **C and D:** Parallel line graph of the evaluation metrics for the top five models. **E and F:** calibration curve analysis; **G and H:** Decision curves analysis of the top five models.
(TIF)

## Author contributions

**Conceptualization:** Yanyan Li.

**Data curation:** Shengtao Li, Ruqiao Luo, Yanfen Li, Aoli Shi.

**Formal analysis:** Zhanzhan Li, Aoli Shi.

**Funding acquisition:** Yanyan Li.

**Investigation:** Shengtao Li, Ruqiao Luo, Yanfen Li, Aoli Shi.

**Methodology:** Zhanzhan Li, Yanfen Li.

**Project administration:** Yanyan Li.

**Resources:** Aoli Shi.

**Software:** Zhanzhan Li.

**Validation:** Zhanzhan Li.

**Visualization:** Zhanzhan Li, Ruqiao Luo.

**Writing – original draft:** Shengtao Li.

**Writing – review & editing:** Zhanzhan Li, Ruqiao Luo, Yanfen Li, Aoli Shi, Yanyan Li.

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
