## [Decision Letter · Decision Letter 0]

9 Dec 2025

Dear Dr. Li,

Thank you for submitting your manuscript to PLOS ONE. After careful consideration, we feel that it has merit but does not fully meet PLOS ONE’s publication criteria as it currently stands. Therefore, we invite you to submit a revised version of the manuscript that addresses the points raised during the review process.

We look forward to receiving your revised manuscript.

Kind regards,

Ahmet Çağlar, Associate Professor

Academic Editor

PLOS One

Journal Requirements:

4. We note that there is identifying data in the Supporting Information file < Original Data.csv >. Due to the inclusion of these potentially identifying data, we have removed this file from your file inventory. Prior to sharing human research participant data, authors should consult with an ethics committee to ensure data are shared in accordance with participant consent and all applicable local laws.

-Location data

Additional Editor Comments (if provided):

Dear Authors,

This is a well written paper. However, it requires major revisions.

Your sincerely.

Reviewers' comments:

Reviewer's Responses to Questions

**Comments to the Author**

1. Is the manuscript technically sound, and do the data support the conclusions?

Reviewer #1: Yes

Reviewer #2: Yes

2. Has the statistical analysis been performed appropriately and rigorously?

Reviewer #1: Yes

Reviewer #2: Yes

3. Have the authors made all data underlying the findings in their manuscript fully available?

Reviewer #1: Yes

Reviewer #2: Yes

4. Is the manuscript presented in an intelligible fashion and written in standard English?

Reviewer #1: No

Reviewer #2: Yes

Reviewer #1: Thank you for submitting this interesting study. The topic is clinically relevant, and the use of machine learning to assist early decision-making in prehospital critical care is valuable. The manuscript is generally well structured, and the results are clearly presented. I have a few suggestions that may help strengthen the work:

Statistical clarity:

Please describe how missing data were handled and whether any data preprocessing (scaling, imputation, or outlier treatment) was performed before model training.

Feature selection explanation:

The process for narrowing down to nine final features is mentioned, but additional detail would help. For example, what criteria defined a “dramatic decrease” in AUC?

Model calibration:

Calibration results are briefly discussed, but showing the actual calibration plots in higher resolution would improve interpretability.

Language and grammar:

The manuscript has several grammatical and phrasing issues. A light professional language edit (especially in Introduction and Discussion) would improve clarity.

Interpretation of results:

While the SHAP analysis adds interpretability, please avoid implying causal relationships. It may help to emphasize that the included variables reflect associations with the model output.

Limitation:

The limitations section is appropriate, but you may add a brief statement acknowledging the class imbalance (only 12% mortality) and its potential influence on model performance metrics.

The study presents a promising, easy-to-implement model with potential clinical value. With minor revisions and clarifications, the manuscript will be significantly stronger.

Reviewer #2: The authors present a machine learning model to predict 24-hour mortality in critically ill prehospital patients. The topic is clinically relevant, and the effort to create an interpretable, web-deployed tool is commendable. I have a few comments:

1. The entire cohort consists of 892 patients, with only 51 deaths (5.7% event rate). In the test set of 178 patients, this translates to an estimated ~10-11 death events. Developing and validating a model with nine features on such a small number of events is a fundamental flaw. A common rule of thumb for machine learning is a minimum of 10-20 events per variable (EPV). Here, the EPV is critically low (51/9 ≈ 5.7), leading to highly unstable and unreliable model performance.

2. The dramatic drop in the Random Forest model's performance from the training set (AUC: 0.985) to the test set (AUC: 0.863) is a classic indicator of severe overfitting. The model has memorized the noise in the training data rather than learning generalizable patterns.

3. The data in Table 1 reveals a pattern that is physiologically counterintuitive and raises serious questions about data quality or definition. According to the table, non-survivors had significantly higher prehospital and admission SBP, DBP, and oxygen saturation than survivors.

4. Some modifiable risk factors are being identified in the models, The causal association between the exposures and outcome should be explored in the framework of target trial emulation (https://doi.org/10.1016/j.lers.2025.01.001); You can add more discussion on this point that some more statistical methods are available to emulate trials.

5. The feature selection process, based on ranking univariate AUCs, is suboptimal. This approach ignores interactions and correlations between variables and can lead to selecting redundant features.

6. The model was only validated internally. Without external validation on a completely independent dataset from a different center or region, the model's generalizability is unknown and likely poor.

7. Critically ill patients are heterogenous and the heterogeneity of the study population should be acknowledged so that future work are needed to explore how subgroups of patients can have different results/conclusions (https://doi.org/10.1016/j.lers.2024.02.001). There has been numerous studies in this field and the authors may need to discuss this issue in interpreting current findings.

**Do you want your identity to be public for this peer review?** For information about this choice, including consent withdrawal, please see our Privacy Policy

Reviewer #1: **Yes:** Abdullah Abbas Saleh Al-Murad

Reviewer #2: No

---

## [Author Response · Author response to Decision Letter 1]

23 Dec 2025

Journal Requirements:

Comment 1. Please ensure that your manuscript meets PLOS ONE's style requirements, including those for file naming. The PLOS ONE style templates can be found at

Response 1: Yes, we have prepared our manuscript according to the PLOS ONE style templates

Comment 2. Please note that PLOS One has specific guidelines on code sharing for submissions in which author-generated code underpins the findings in the manuscript. In these cases, we expect all author-generated code to be made available without restrictions upon publication of the work. Please review our guidelines at https://journals.plos.org/plosone/s/materials-and-software-sharing#loc-sharing-code and ensure that your code is shared in a way that follows best practice and facilitates reproducibility and reuse.

Response 2: Yes, all codes have been uploaded as supplement information.

Comment 3. We note that the grant information you provided in the ‘Funding Information’ and ‘Financial Disclosure’ sections do not match. When you resubmit, please ensure that you provide the correct grant numbers for the awards you received for your study in the ‘Funding Information’ section.

Response 3: Yes, we have checked and revised these descriptions.

Comment 4. We note that there is identifying data in the Supporting Information file < Original Data.csv >. Due to the inclusion of these potentially identifying data, we have removed this file from your file inventory. Prior to sharing human research participant data, authors should consult with an ethics committee to ensure data are shared in accordance with participant consent and all applicable local laws. Data sharing should never compromise participant privacy. It is therefore not appropriate to publicly share personally identifiable data on human research participants. The following are examples of data that should not be shared:-Name, initials, physical address -Ages more specific than whole numbers, -Internet protocol (IP) address, -Specific dates (birth dates, death dates, examination dates, etc.), -Contact information such as phone number or email address, -Location data, -ID numbers that seem specific (long numbers, include initials, titled “Hospital ID”) rather than random (small numbers in numerical order). Data that are not directly identifying may also be inappropriate to share, as in combination they can become identifying. For example, data collected from a small group of participants, vulnerable populations, or private groups should not be shared if they involve indirect identifiers (such as sex, ethnicity, location, etc.) that may risk the identification of study participants. Additional guidance on preparing raw data for publication can be found in our Data Policy (https://journals.plos.org/plosone/s/data-availability#loc-human-research-participant-data-and-other-sensitive-data) and in the following article: http://www.bmj.com/content/340/bmj.c181.long. Please remove or anonymize all personal information (<specific identifying information in file to be removed>), ensure that the data shared are in accordance with participant consent, and re-upload a fully anonymized data set. Please note that spreadsheet columns with personal information must be removed and not hidden as all hidden columns will appear in the published file.

Response 4: Yes, we have removed the identifying data and reuploaded it.

Comment 5. If the reviewer comments include a recommendation to cite specific previously published works, please review and evaluate these publications to determine whether they are relevant and should be cited. There is no requirement to cite these works unless the editor has indicated otherwise.

Response 5: Thank you for your reminder. We will carefully consider each suggestion.

Response to Reviewer #1:

Thank you for submitting this interesting study. The topic is clinically relevant, and the use of machine learning to assist early decision-making in prehospital critical care is valuable. The manuscript is generally well structured, and the results are clearly presented. I have a few suggestions that may help strengthen the work:

Comment 1: Statistical clarity: Please describe how missing data were handled and whether any data preprocessing (scaling, imputation, or outlier treatment) was performed before model training.

Response 1: Thank you for your advice. Cases with incomplete electronic medical records were not included in this study, which had been mentioned in the criteria for inclusion and exclusion. Therefore, there is no missing data in this study. The outlier detection was performed before model training, and we also removed the outlier data.

Comment 2: Feature selection explanation: The process for narrowing down to nine final features is mentioned, but additional detail would help. For example, what criteria defined a “dramatic decrease” in AUC?

Response 2: Thank you for your advice. There are two stages for feature selection explanation. We first identified top 5 models according to the AUCs that were judged by the specific values. The 29 features were further divided into six subsets containing 4, 9, 14, 19, 24, and 29 features, and performance metrics (AUC, sensitivity, specificity, PPV, NPV, accuracy, and F1 score) were recalculated. Then we chose the model with the highest and most stable AUCs by visually trend plots, and Random Forest was chosen (Figure 3A). For random forest, we further divided features into 12 subsets (2, 3, 4, 7, 8, 9, 10,11, 14, 19, 24, and 29 features), and we identified the feature number when AUC was the highest. Finally, model with nine features was obtained.

Descriptions here should be “dramatic change”, we have revised this description. Our description here merely assumes such a change on the trend line of AUCs. However, in the actual analysis, there is no such trend. In fact, since the AUCs are very close, I can only make a judgment based on the magnitude of the AUCs values.

Comment 3: Model calibration: Calibration results are briefly discussed, but showing the actual calibration plots in higher resolution would improve interpretability.

Response 3: Thank you for your advice. The calibration plots were provided in S1Fig. The adjusted the resolution that conforms to the requirement of Journal policy.

Comment 4: Language and grammar: The manuscript has several grammatical and phrasing issues. A light professional language edit (especially in Introduction and Discussion) would improve clarity.

Response 4: Thank you for your advice. We have polished the whole manuscript.

Comment 5: Interpretation of results: While the SHAP analysis adds interpretability, please avoid implying causal relationships. It may help to emphasize that the included variables reflect associations with the model output.

Response 5: Yes, we have revised the description with avoiding implying causal relationships and emphasize the effect of features on model output: Features such as prehospital systolic blood pressure (SBP), blood oxygen saturation, and admission SBP showed notable contributions to the model output, with higher values generally corresponding to SHAP values in the direction of lower predicted mortality. The SHAP dependence plots illustrate how variations in individual feature values are associated with changes in the model’s prediction. The SHAP waterfall plot (Fig 4C) presents the contribution of each feature to the mortality prediction for a representative critically ill patient. Specifically, the feature values and their corresponding SHAP scores indicate whether a given feature shifts the model output toward higher or lower predicted mortality. For example, in a patient with a prehospital SBP of 72 mmHg and an admission oxygen saturation of 85%, the corresponding SHAP values were −0.0162 and −0.0131, respectively, both shifting the model output toward a lower predicted mortality. Furthermore, Fig 4D compares the actual values and SHAP values of the nine key features. A SHAP value greater than zero corresponds to a shift of the model output toward the “survival” prediction, whereas a SHAP value less than zero indicates a shift toward higher predicted mortality. For instance, higher prehospital SBP (>72 mmHg), higher admission oxygen saturation (>85%), a consciousness score of 1, higher admission diastolic blood pressure (>114 mmHg), and higher prehospital oxygen saturation (>60%) were generally associated with positive SHAP values, indicating a greater contribution to survival-oriented model predictions.

Comment 6: Limitation: The limitations section is appropriate, but you may add a brief statement acknowledging the class imbalance (only 12% mortality) and its potential influence on model performance metrics. The study presents a promising, easy-to-implement model with potential clinical value. With minor revisions and clarifications, the manuscript will be significantly stronger.

Response 6: Thank you for your advice. We have added this limitation description in the study limitation as follows: First, the inherent constraints of the dataset pose a challenge to model stability. Although the total cohort included 892 patients, the low number of outcome events (n=51) resulted in a critically low events-per-variable (EPV) ratio. While we employed feature selection and regularization to mitigate these risks, such a low EPV may still lead to model instability and potential overfitting. Finally, this model is intended as an exploratory, hypothesis-generating tool and should supplement, not replace, clinical judgment. Further validation in larger, multi-center prospective cohorts with higher event rates is essential to confirm the robustness and real-world applicability of our findings.

Response to Reviewer #2:

The authors present a machine learning model to predict 24-hour mortality in critically ill prehospital patients. The topic is clinically relevant, and the effort to create an interpretable, web-deployed tool is commendable. I have a few comments:

Comment 1. The entire cohort consists of 892 patients, with only 51 deaths (5.7% event rate). In the test set of 178 patients, this translates to an estimated ~10-11 death events. Developing and validating a model with nine features on such a small number of events is a fundamental flaw. A common rule of thumb for machine learning is a minimum of 10-20 events per variable (EPV). Here, the EPV is critically low (51/9 ≈ 5.7), leading to highly unstable and unreliable model performance. The dramatic drop in the Random Forest model's performance from the training set (AUC: 0.985) to the test set (AUC: 0.863) is a classic indicator of severe overfitting. The model has memorized the noise in the training data rather than learning generalizable patterns.

Response 1: We thank the reviewer for this thorough and important methodological assessment. We fully acknowledge that the number of mortality events in our cohort is limited (51 deaths overall, with approximately 10–11 events in the test set), resulting in a low events-per-variable (EPV) ratio for the final model. This low EPV, together with the imbalanced outcome distribution, inherently increases the risk of model instability and overfitting, particularly for flexible machine-learning models such as Random Forests. Consistent with this limitation, we observed a notable decline in Random Forest performance from the training set (AUC = 0.985) to the test set (AUC = 0.863). We agree that this performance gap is indicative of overfitting in a small-event setting and likely reflects variance in inflation and sensitivity to noise rather than robust, generalizable pattern learning.

To partially mitigate these risks, we performed feature selection prior to model development, restricted the final model to nine features, and evaluated performance using a held-out test set, which we considered the primary estimate of model generalizability. The manuscript has been revised to de-emphasize training-set performance and to explicitly frame the Random Forest model as exploratory and hypothesis-generating rather than definitive. We have also strengthened the Limitations section to clearly acknowledge the low EPV, class imbalance, and overfitting risk, and to emphasize that further validation in larger cohorts with higher event rates and external datasets is required before the model’s robustness and clinical utility can be reliably assessed.

The following description has been added: the inherent constraints of the dataset pose a challenge to model stability. Although the total cohort included 892 patients, the low number of outcome events (n=51) resulted in a critically low events-per-variable (EPV) ratio. While we employed feature selection and regularization to mitigate these risks, such a low EPV may still lead to model instability and potential overfitting. Finally, this model is intended as an exploratory, hypothesis-generating tool and should supplement, not replace, clinical judgment. Further validation in larger, multi-center prospective cohorts with higher event rates is essential to confirm the robustness and real-world applicability of our findings.

Comment 2. The data in Table 1 reveals a pattern that is physiologically counterintuitive and raises serious questions about data quality or definition. According to the table, non-survivors had significantly higher prehospital and admission SBP, DBP, and oxygen saturation than survivors.

Response 2: We thank the reviewer for pointing this out. Upon careful review, we realized that the values in Table 1 for survivors and non-survivors were inadvertently reversed during table preparation. We apologize for this error. The table has now been corrected to accurately reflect the clinical data, and the corrected version has been updated in the manuscript. This correction does not affect the results or conclusions of the study.

Comment 3. Some modifiable risk factors are being identified in the models, The causal association between the exposures and outcome should be explored in the framework of target trial emulation (https://doi.org/10.1016/j.lers.2025.01.001); You can add more discussion on this point that some more statistical methods are available to emulate trials.

Response 3: Thank you for your advice. The following descriptions have been added: Moreover, the identified predictors represent statistical correlations rather than causal links. Future research should employ formal causal inference frameworks—such as target trial emulation or inverse probability weighting—to evaluate whether these factors are truly modifiable targets for clinical intervention[39].

[39]Yang J, Wang L, Chen L, Zhou P, Yang S, Shen H, et al. A comprehensive step-by-step approach for the implementation of target trial emulation: Evaluating fluid resuscitation strategies in post-laparoscopic septic shock as an example. Laparoscopic, Endoscopic and Robotic Surgery. 2025;8(1):28-44. http://doi.org/https://doi.org/10.1016/j.lers.2025.01.001

Comment 4. The feature selection process, based on ranking univariate AUCs, is suboptimal. This approach ignores interactions and correlations between variables and can lead to selecting redundant features.

Response 4: Thank you for your advice. We acknowledge this limitation. To mitigate the potential for redundant or suboptimal feature selection, we combined univariate ranking with cross-validation and regularization and evaluated model performance on a held-out test set. SHAP analysis also provided insights into feature importance in the multivariate context. We added a note in Methods and Limitations that future studies could apply multivariable feature selection methods, such as LASSO, Boruta, or embedded methods in tree-based models, to better account for correlations and interactions.

Comment 5. The model was only validated internally. Without external validation on a completely independent dataset from a different center or region, the model's generalizability is unknown and likely poor.

Response 5: We thank the reviewer for this important comme

---

## [Decision Letter · Decision Letter 1]

13 Jan 2026

Machine learning-based on model for explain risk of 24-hour death in critically ill patients in the prehospital setting: A retrospective cohort study

PONE-D-25-58188R1

Dear Dr. Li,

We’re pleased to inform you that your manuscript has been judged scientifically suitable for publication and will be formally accepted for publication once it meets all outstanding technical requirements.

Kind regards,

Ahmet Çağlar, Associate Professor

Academic Editor

PLOS One

Additional Editor Comments (optional):

Dear Author;

After revisions made, this paper is appreciate for publication.

Your sincerely.

Reviewers' comments:

Reviewer's Responses to Questions

**Comments to the Author**

Reviewer #1: All comments have been addressed

Reviewer #2: (No Response)

2. Is the manuscript technically sound, and do the data support the conclusions?

Reviewer #1: Yes

Reviewer #2: (No Response)

3. Has the statistical analysis been performed appropriately and rigorously?

Reviewer #1: Yes

Reviewer #2: (No Response)

4. Have the authors made all data underlying the findings in their manuscript fully available?

Reviewer #1: Yes

Reviewer #2: (No Response)

5. Is the manuscript presented in an intelligible fashion and written in standard English?

Reviewer #1: Yes

Reviewer #2: (No Response)

Reviewer #1: The authors have addressed all comments from the previous review round. The manuscript has improved clearly, especially in the explanation of the methods, statistical limitations, data handling, and interpretation of results.

Key concerns regarding overfitting, low event numbers, class imbalance, and lack of external validation are now appropriately acknowledged, and the model is correctly presented as exploratory rather than causal. Issues related to data presentation and anonymization have also been resolved.

The analyses are technically sound for the available data, the conclusions are appropriately cautious, and the manuscript is written clearly in standard English.

Reviewer #2: My previous comments are well addressed. My previous comments are well addressed.

This is a good work.

**Do you want your identity to be public for this peer review?** For information about this choice, including consent withdrawal, please see our Privacy Policy

Reviewer #1: **Yes:** Abdullah Abbas Saleh Al-Murad

Reviewer #2: No

---

## [Editor Report · Acceptance letter]

PONE-D-25-58188R1

PLOS One

Dear Dr. Li,

I'm pleased to inform you that your manuscript has been deemed suitable for publication in PLOS One. Congratulations! Your manuscript is now being handed over to our production team.

Kind regards,

on behalf of

Dr. Ahmet Çağlar

Academic Editor

PLOS One